# How Changes in the Nutritional Landscape Shape Gut Immunometabolism

**DOI:** 10.3390/nu13030823

**Published:** 2021-03-02

**Authors:** Jian Tan, Duan Ni, Rosilene V. Ribeiro, Gabriela V. Pinget, Laurence Macia

**Affiliations:** 1The Charles Perkins Centre, The University of Sydney, Sydney, NSW 2006, Australia; jian.tan@sydney.edu.au (J.T.); duan.ni@sydney.edu.au (D.N.); rosie.ribeiro@sydney.edu.au (R.V.R.); gabriela.pinget@sydney.edu.au (G.V.P.); 2School of Medical Sciences, Faculty of Medicine and Health, The University of Sydney, Sydney, NSW 2006, Australia; 3School of Life and Environmental Science, Faculty of Science, The University of Sydney, Sydney, NSW 2006, Australia

**Keywords:** immunometabolism, gut immunity, nutrition, gut microbiota

## Abstract

Cell survival, proliferation and function are energy-demanding processes, fuelled by different metabolic pathways. Immune cells like any other cells will adapt their energy production to their function with specific metabolic pathways characteristic of resting, inflammatory or anti-inflammatory cells. This concept of immunometabolism is revolutionising the field of immunology, opening the gates for novel therapeutic approaches aimed at altering immune responses through immune metabolic manipulations. The first part of this review will give an extensive overview on the metabolic pathways used by immune cells. Diet is a major source of energy, providing substrates to fuel these different metabolic pathways. Protein, lipid and carbohydrate composition as well as food additives can thus shape the immune response particularly in the gut, the first immune point of contact with food antigens and gastrointestinal tract pathogens. How diet composition might affect gut immunometabolism and its impact on diseases will also be discussed. Finally, the food ingested by the host is also a source of energy for the micro-organisms inhabiting the gut lumen particularly in the colon. The by-products released through the processing of specific nutrients by gut bacteria also influence immune cell activity and differentiation. How bacterial metabolites influence gut immunometabolism will be covered in the third part of this review. This notion of immunometabolism and immune function is recent and a deeper understanding of how lifestyle might influence gut immunometabolism is key to prevent or treat diseases.

## 1. Introduction

Animal diets are comprised of a complex mixture of macro- and micro-nutrients, which provide a vital source of energy or act as catalytic cofactors necessary to maintain cellular function. Carbohydrate, protein and fat are described as macronutrients as they are required in large quantities (grams) while micronutrients - vitamins and minerals - are needed in low amounts (micro- to milli-grams). The current recommendation of macronutrients for humans is 20–35% of total energy intake from fat, 45–65% from carbohydrate and 15–25% from protein [1]. Apart from their importance as an energy source for cells, macronutrients are also the staple for the production of macromolecules such as proteins, cholesterol and nucleic acids. Macronutrients are not homogenous, with food containing different types of carbohydrates, lipids and proteins. Carbohydrates are classified as simple or complex depending on their digestibility: simple carbohydrates are quickly digested into monosaccharides by host enzymes whereas complex carbohydrates (containing three or more sugars bonded together in a complex structure) take longer to be digested. Dietary fibre and resistant starch are examples of complex carbohydrates that can only be digested with the assistance of the host’s gut bacteria. Lipids are comprised of medium- and long-chain fatty acids depending on their carbon length and can be further classified as saturated, mono- or poly-unsaturated depending on the presence of zero, one or multiple double bonds. Proteins are the source of amino acids and are classified as essential, which can only originate from dietary sources, or non-essential which can be produced by the organism. Vitamins are key micronutrients and are classified as either fat-soluble (vitamins A, D, E and K), which can be stored in the liver, or water-soluble (vitamins B and C), which excess are excreted in the feces and urine and therefore must be replenished continuously. Along with minerals such as magnesium, zinc and iron, vitamins serve as cofactors for enzymes involved in metabolism of macronutrients, such as those involved in glycolysis and fatty acid oxidation [2]. Vitamins can also be directly metabolized to produce immunomodulatory products.

Like any other cells, the development and function of immune cells rely on the type and amount of food consumed. Both over- and under-nutrition, including micronutrient deficiency, are linked to poor immune outcomes. Undernutrition is linked to greater morbidity and mortality from infections [3], while obesity is linked to increased infection rate, and is associated with immune disorders such as allergies and autoimmunity [4]. While specific defects to the immune system under different nutritional context are yet to be fully defined, it highlights that the nutritional status of the host has a broad impact on immune function [3].

Immune cells can sense nutrients via numerous mechanisms. This can involve intracellular nutrient sensors, such as the mechanistic target of rapamycin (mTOR), and aryl hydrocarbon receptor (AhR) that binds ligands derived from cruciferous vegetables, or metabolite-sensing G protein-coupled receptor (GPCR) that binds to a variety of host- and microbial-derived metabolites [5]. For example, short-chain fatty acids produced under high fibre feeding condition can bind to specific GPCR expressed on immune cell to alter their phenotype. These metabolites might also directly fuel specific metabolic pathways to affect immune cell activity.

While the energy demanding aspect of the immune response is recognized, it is only recently that the impact of metabolic activity on immune cell function has been appreciated. The type of energetic substrates available to an immune cell will activate specific metabolic pathways, which can dictate whether it becomes pro- or anti-inflammatory. Thus, both the type and amount of food consumed have the potential to affect immunometabolism, and there is potential for diet to be used as a tool to fine tune the immune response. In this review, we will give an overview on how different sources of dietary components (including food additives) can affect immune cell metabolism (immunometabolism) to drive their differentiation and function, with particular focus on how macro- and micro-nutrient affect gut immunity.

## 2. Immunometabolism and Impact on Immune Cell Differentiation and Function

Immune cells have different energy requirements depending on their activation state. Naïve T cells and resting regulatory T cells (Treg) are maintained in a quiescent state and require low cellular metabolic and biosynthetic activity to maintain their survival [6]. In contrast, activated T cells must be metabolically active to meet the energetic demands required for their proliferation and effector functions. The switch from a quiescent to a metabolically active state is associated with the upregulation of nutrient transporters that will in turn activate specific intracellular metabolic pathways to produce adenosine triphosphate (ATP) as an energy source. Of note, high affinity T cells have enhanced nutrient uptake and sustained metabolic activity, and thus outcompete T cells with low-affinity T-cell receptor for nutrient access. This highlight nutrients as another dimension of the immune response [7].

The type of nutrients used for ATP production also differs between cells, with glucose and glutamine preferred by activated B and T lymphocytes while activated natural killer cells utilise only glucose [8,9]. Preferential utilisation of specific metabolic pathways can determine the differentiation of immune cells towards a pro- or anti-inflammatory phenotype. Glycolysis is typically the “pro-inflammatory immuno-metabolic pathway” characteristic of T helper (Th)-1, Th17 cells, pro-inflammatory macrophages (M1) and activated dendritic cells (DC), while fatty acid oxidation and oxidative phosphorylation are typically the “anti-inflammatory immune-metabolic pathways” fuelling Treg and anti-inflammatory macrophages (M2) [9]. The major intracellular metabolic pathways include glycolysis, the pentose phosphate pathway (PPP), beta-oxidation, tricarboxylic acid cycle, amino acid metabolism, mevalonate pathway, ketogenesis and fatty acid metabolism. Their impact on immune cell differentiation and function is detailed below, and schematic diagrams depicting these pathways have been published elsewhere [10,11].

### 2.1. Glycolysis Mediates Proinflammatory Effects

Glycolysis is the major metabolic pathway for the conversion of simple carbohydrates into energy. It is an ubiquitous pathway present in all cell types, including erythrocytes. While the main substrate for glycolysis is glucose, fructose and galactose may also enter this pathway through alternative routes. Glycolysis results in the generation of pyruvate, ATP and reduced nicotinamide adenine dinucleotide (NADH) from glucose. Pyruvate can be subsequently converted into acetyl-CoA in the mitochondria to fuel the tricarboxylic acid cycle (TCA) cycle. The TCA cycle provides a source of mitochondrial NADH and flavin adenine dinucleotide (FADH_2_) required for oxidative phosphorylation (OXPHOS), an oxygen-dependent pathway that ultimately yields 36 ATP from 1 molecule of glucose. Despite this energetic efficiency, it has been observed that proinflammatory immune cells preferentially undergo glycolysis for ATP production even in the presence of oxygen (a process known as aerobic glycolysis). In such cases, pyruvate is mostly redirected to produce lactate in the cytoplasm, resulting in the oxidation of NADH to nicotinamide adenine dinucleotide (NAD^+^) to further sustain glycolysis as the major metabolic machinery for generating ATP. Excess lactate is mostly secreted or can be retained for conversion back into pyruvate [12]. Aerobic glycolysis may be observed when energy is urgently needed as the rate of ATP production from glycolysis is much faster than oxidative phosphorylation [13]. Glycolysis is also required to drive the PPP, which produces intermediates required for the synthesis of fatty acids, cytokines, and nucleotides for cell proliferation. Glycolysis is also relied on by immune cells at sites of inflammation, which are generally hypoxic and where OXPHOS is inhibited [14]. The significance of this was highlighted in a recent study where the knockdown of the glycolytic enzyme glucose phosphate isomerase 1 specifically depleted pathogenic Th17 and not homeostatic Th17 [15]. This was linked to the inability of pathogenic Th17 cells to compensate with the PPP to maintain glycolytic flux under hypoxic conditions [15].

Despite the apparent association of aerobic glycolysis to pro-inflammatory immune phenotypes, lactate appears to have anti-inflammatory effects. Lactate has been shown to dampen the activity of innate immune cells such as monocytes and macrophages, by decreasing NLR family pyrin domain containing 3 (NLRP3) inflammasome activation and subsequent interleukin-1β (IL-1β) production via activation of lactate receptor G protein-coupled receptor (GPR)-81 [16]. Lactate also inhibits pro-inflammatory cytokine production in macrophages by directly inhibiting glycolysis [17]. Lactate has been shown to affect adaptive immunity by skewing the pro-inflammatory Th1/Th17 and anti-inflammatory Treg cell balance towards Treg development with the absence of GPR81 aggravating colitis development in mice [18]. However, on the downside of this anti-inflammatory effect, lactate produced by cancerous cells blocks the anti-tumour response of tumour infiltrating cytolytic CD8^+^ T cells [19].

The requirement for rapid mobilization and immediate response of immune cells to threats such as infectious agents may explain the link between glycolysis and inflammation. The associated production of lactate might be a preserved strategy to control a potentially damaging inflammatory response.

### 2.2. Pro-Inflammatory Effects of the Pentose Phosphate Pathway 

The PPP is an alternative pathway that runs in parallel to glycolysis. It involves the utilization of glycolysis intermediate glucose-6-phosphate (G6P) by the enzyme glucose-6-phosphate dehydrogenase (G6PD), generating the major source of cytosolic NADPH required for fatty acid synthesis, as well as for the reduction of glutathione which plays anti-oxidative roles. The PPP also generates precursors necessary for the production of nucleotide, of α-amino acids, such as histidine and serine, required for cellular proliferation and cytokine production. Alternatively, PPP intermediates can feed back into glycolysis, which is a feature of homeostatic but not pathogenic Th17 cells [15]. PPP is enhanced in lipopolysaccharide (LPS)-stimulated macrophages leading to the production of reactive oxygen species (ROS) through the oxidation of NADPH, and blockade of G6PD was shown to decrease LPS-induced tumor necrosis factor (TNF) and IL-6 production [20]. Accordingly, the inhibition of PPP under hypercholesterolemic conditions decrease the inflammatory profile of macrophages [21]. Similarly, PPP is necessary for neutrophil microbicidal activity through increased ROS production and G6PD was necessary for neutrophil extracellular trap formation [22]. Overall, the PPP appears to be a key metabolic pathway that supports the pro-inflammatory activity of effector immune cells.

### 2.3. Gluconeogenesis Drives the Effector Function of Immune Cells by Supporting Glycogenolysis

Unlike glycolysis, gluconeogenesis is a metabolic pathway that results in the generation of glucose. The major substrates for gluconeogenesis are lactate, glycerol (derived from triglyceride) and glucogenic amino acids. Within immune cells, gluconeogenesis is used to drive glycogenesis, a process by which intracellular glycogen stores are generated. Glutamine-mediated gluconeogenesis and glycogenesis was shown to be important for neutrophil effector function as well as for their survival during an immune response, by providing a source of glucose from glycogen through glycogenolysis [23]. Upregulation of both gluconeogenesis and glycogenolysis is also a feature of CD8^+^ memory T cells where glycogen was shown to be important for their formation and maintenance with glucose being converted into G6P fuelling the PPP [24]. Glycogenolysis also supports DC survival, maturation, and effector function and G6P derived from glycogen metabolism performed different intracellular functions to G6P derived from glucose [25]. In macrophages, glycogenolysis was also used to drive the PPP to support inflammatory macrophage survival as well as maintaining the inflammatory phenotype of macrophages through uridine diphosphate glucose-P2Y purinoceptor 14 signalling [26]. Thus, gluconeogenesis appears to drive proinflammatory immune effector function.

### 2.4. Anti-Inflammatory Effects of Fatty Acid Metabolism by Beta Oxidation 

The other major pathway for ATP generation is through the oxidation of fatty acids via beta-oxidation. Cytosolic medium- and long-chain fatty acids are first converted into acyl-CoA molecules by long-chain fatty acyl-CoA synthetase and subsequently shuttled into the mitochondria by the enzymes carnitine palmitoyltransferase (CPT), carnitine translocase and CPT-2. The acyl-CoA chain then undergoes repeated cycles of oxidation (2 carbon length at a time), yielding 2 molecules of acetyl-CoA at each cycle. The oxidation of very-long-chain fatty acids predominantly occurs in cytosolic peroxisomes to generate shorter-chained acyl-CoA that can be subsequently shuttled to the mitochondrial to be further oxidised.

Beta-oxidation is typically associated with an anti-inflammatory immune phenotype and is a metabolic pathway predominant in Treg and M2 macrophages [27]. Pharmacological inhibition of this pathway using the CPT-1 inhibitor etomoxir impaired the differentiation of both Tregs and M2 macrophages yet the specific deletion of *Cpt1a* in T cells did not alter its potential to differentiate into Tregs and of *Cpt2* did not impair M2 macrophage differentiation [28]. Fatty acid oxidation is also crucial to generate CD8^+^ memory T cell, but not effector T cell following infection or vaccination, through TNF receptor-associated factor 6 receptor-dependent mechanisms [29]. As before, specific deletion of *Cpt1a* in T cells did not alter its potential to differentiate into memory T cells [30]. Thus, the reliance on beta-oxidation by macrophages and T cells do not appear to drive their differentiation but likely support their maintenance or functionality. A defect in beta-oxidation also results in the accumulation of acyl-L-carnitine, which promotes IL-17 cytokine production and thus differentiation of T cells towards Th17 cells [31].

However, the association of beta-oxidation to anti-inflammatory phenotypes is not an obligate rule. Under glucose-limited conditions, neutrophil effector function can be rescued by fatty acid oxidation [32], and the inhibition of mitochondrial lipid transporter FABP5 enhanced Treg suppressive capability by promoting IL-10 production [33]. Furthermore, highly proliferative germinal centre B cells preferentially generate energy from fatty acids oxidation over glycolysis [34]. This is surprising as such conditions result in a hypoxic microenvironment that should limit the capacity for beta-oxidation and OXPHOS. Overall, fatty acid oxidation is mostly associated with anti-inflammatory effects but can promote pro-inflammatory cells under specific conditions.

### 2.5. Pro-Inflammatory Effects of Fatty Acid Synthesis and Lipogenesis 

Fatty acid synthesis occurs in the cytoplasm via the action of fatty acid synthase, which synthesize palmitic acid from citrate-derived Acetyl-CoA. A key regulatory step is the irreversible conversion of acetyl-CoA to malonyl-CoA by Acetyl-CoA carboxylase. Fatty acid synthesis also requires cytosol NADPH generated from the PPP. This pathway is typically linked to a pro-inflammatory immune phenotype, possibly due to its association with glycolysis and the PPP. Blockade of fatty acid synthesis in T cell through the specific deletion Acetyl-CoA carboxylase blocked the differentiation of Th17 cells while promoting Treg differentiation [35]. *De novo* synthesis of oleic acid by B cells was essential for maintaining B cell fitness by supporting OXPHOS and mTOR activity [36]. *De novo* lipogenesis has been reported in LPS-stimulated B cells, which was shown to support their proliferation and production of antibodies [37]. Finally, fatty acid synthesis also drives innate cell towards a pro-inflammatory phenotype, by promoting M1 macrophage differentiation as well as by promoting neutrophil survival during inflammation [38].

### 2.6. The Tricarboxylic Acid Cycle (TCA) and Its Intermediates Have either Pro- or Anti-Inflammatory Effects

Considered as the central metabolic hub, the TCA cycle (or the citric acid cycle) bridges all major metabolic pathways within the cell. The main function of the TCA cycle is to generate ATP, which involves the oxidation of acetyl-CoA to drive oxidative phosphorylation in the mitochondria. The first step of the TCA cycle involves the generation of citrate from Acetyl-CoA and oxaloacetate. Citrate is subsequently isomerized to iso-citrate by the enzyme aconitase, producing cis-aconitate as an intermediate. Isocitrate is then oxidized to α-ketoglutarate which is then subsequently oxidized to form succinyl-CoA. Succinyl-CoA is hydrolysed to generate succinate, and further successive enzymatic reactions result in the generation of fumarate, malate and the eventual regeneration of oxaloacetate. NADH and FADH_2_ produced by the TCA cycle can be oxidized within the electron transport chain (ETC) to generate an electrochemical H^+^ gradient to drive ATP production via the action of ATP synthase. Other than driving ATP production, TCA cycle intermediates can be used for the synthesis of amino acids, fatty acids, nucleotides, and other macromolecules.

The conversion of citrate to cis-aconitate and subsequently into itaconate has been shown to have anti-inflammatory effects in macrophages by inhibiting succinate dehydrogenase (SDH), which in turn downregulated the expression of NO synthase, IL-1β, IL-18, and hypoxia-inducible factor 1-alpha (HIF-1α) [39,40]. On the contrary, succinate dehydrogenase activity supports proinflammatory macrophage by increasing ROS production [41]. The TCA cycle has two potential breakpoints, at the level of isocitrate dehydrogenase and at the level of succinate, which has been reported in M1 macrophages. Decreased isocitrate dehydrogenase activity leads to the accumulation of citrate that is redirected into itaconate and fatty acid synthesis and the accumulation of malate due to the inefficient conversion of succinate promoted nitric oxide (NO) production by enhancing the arginosuccinate shunt [42]. TCA cycle intermediates and enzymes can thus affect immune cell inflammatory profile and these effects are either through epigenetic changes, post-translational modification or activation of specific G-protein coupled receptors.

#### 2.6.1. Immunomodulation through Epigenetic Changes

The accumulation of succinate (and fumarate) has been shown to alter immune cell histone methylation state by directly inhibiting DNA methylases, resulting in DNA hypermethylation [43]. Altered DNA methylation affects gene expression and can alter the differentiation of numerous immune cell subsets, as well as their function including phagocytosis, cytokine production and their effector response [44]. Succinate and fumarate levels are higher in β-glucan trained monocytes and macrophages, which is associated with epigenetic changes [45]. Succinate can also stabilise HIF-1α by inhibiting prolyl hydroxylases to support glycolysis and production of pro-inflammatory IL-1β, which is inhibited by the glucose analog 2-deoxy-D-glucose [46].

#### 2.6.2. Immunomodulation through Post-Translational Changes

Citrate can be transported to the cytoplasm and converted into acetyl-CoA by the enzyme ATP–citrate lyase. Acetyl-CoA can be used as a substrate for lysine-acetylation of cytoplasmic proteins to regulate their enzymatic activity [47]. For example, post-translational acetylation of glyceraldehyde 3-phosphate dehydrogenase (GAPDH) enhances its activity in CD8^+^ T cells to promote glycolysis and favouring their differentiation towards memory cells [48], while post-translational acetylation of acetyl-CoA synthetase and long-chain acyl dehydrogenase inhibits their activity [49]. Depending on the protein targeted, post-translational acetylation can have pro- or anti-inflammatory effects. The post-translational acetylation of p53 in macrophages promotes pro-inflammatory M1 macrophage polarization [50], while the acetylation of microtubule supports their production of anti-inflammatory cytokine IL-10 [51]. Post-translational acetylation of prostacyclin synthase also promotes Treg differentiation [52]. Acetyl-CoA is also a substrate for nuclear histone protein acetylation mediated by histone acetyltransferases. Histone acetylation mediates epigenetic remodelling by enhancing transcriptional activation and gene expression, and is a key mechanism in the regulation of immune cell differentiation, cytokine production and survival [53,54].

#### 2.6.3. Immunomodulation through G Protein-Coupled Receptor Signalling

Other than modulating immune cell activity via intracellular effects, TCA intermediates can also have extracellular effects through their binding to G-protein coupled receptors. To date, two GPCR has been identified to bind TCA intermediates, GPR91 and GPR99, which binds to succinate and α-ketoglutarate respectively. GPR91 is highly expressed on macrophages and DC [55] while GPR99 is not expressed by immune cells [5]. The impact of GPR91 on immune cell activation seems context dependent. Absence of GPR91 in macrophages exacerbates their inflammatory profile under diet-induced obesity conditions [56] while another study reported that its absence reduced macrophage activation and IL-1β production to reduce rheumatoid arthritis severity [57]. Activation of GPR91 has also been shown to promote anti-inflammatory M2 macrophage differentiation [56], which is aligned with the fact that succinate in drinking water enhanced type 2 immunity and anti-helminth response via activation of GPR91 by tuft cells [58]. GPR91 also regulates dendritic cell activity increasing their migration potential, pro-inflammatory cytokine production as well as their antigen presentation capacity to increase antigen-specific T cell response [55]. Thus, GPR91 does not have a clear pro- or anti-inflammatory effect, but the activation of GPR91 might be a strategy to counterbalance the effects of intracellular succinate.

### 2.7. Amino Acid Metabolism and the GABA Pathway Mediates Both Pro- and Anti-Inflammatory Effects

Amino acids are both produced and utilized in the TCA cycle. They are vital for immune cell maintenance, proliferation and cytokine production and mediate these effects by increasing ATP production, modulating protein activity through post-translational changes, controlling redox balance or by affecting gene expression through epigenetic changes. *De novo* synthesis of certain amino acids is driven by the TCA intermediates alpha-ketoglutarate (glutamate for the synthesis of glutamine, proline and arginine) and oxaloacetate (asparagine and aspartate).

Several amino acids including arginine, glutamine, serine, tryptophan, as well as the branched-chain amino acids leucine, isoleucine and valine have specific impact on immune cells [59]. For example, tryptophan is predominantly metabolised via the kynurenine pathway by indoleamine 2,3-dioxygenase (IDO) as the first step. The resulting end-products are ligands for the nuclear receptor aryl hydrocarbon receptor (AhR), which directly polarizes naïve T cells into anti-inflammatory regulatory T cells [60,61]. In contrast, the inhibition of IDO in Treg skew their phenotype towards a pro-inflammatory Th17-like phenotype [62]. The expression of IDO can directly induce the development of gut anti-inflammatory Tregs, or directly by promoting the activity of tolerogenic mucosal CD103^+^ dendritic cells [63].

Arginine is another key amino acid and can play either anti- or pro-inflammatory roles depending on its metabolic fate. It can be catalysed by inducible nitric oxide synthase to generate citrulline and nitric oxide, which mediates ROS production in M1 macrophages. Alternatively, arginine can be catalysed by arginase to generate ornithine and urea, a feature of M2 macrophages. Ornithine is a precursor for putrescine and L-proline, required for cell proliferation and collagen synthesis contributing to wound healing. Arginine is also a nutrient signal that can indirectly promote mTOR signalling by binding to Castor1 and decreased mTOR activity is reported in T cells under arginine-depleted conditions [64]. It can induce immune cell metabolic switch towards OXPHOS to enhance activated T cell survival and cytolytic activity [65].

Glutamine is the most abundant amino acid and can fuel the TCA cycle by its conversion to α-ketoglutarate via glutaminolysis. Intermediates of glutaminolysis can differentially regulate immune phenotype, with uridine diphosphate N-acetylglucosamine promoting M2 macrophage polarization while succinate promotes M1 macrophage polarization [66]. Adaptive immune cells also rely on glutaminolysis to promote germinal center B cell activation, T cell activation, cytokine production [67] and survival by up-regulating glutathione and Bcl-2 [68]. Glutaminolysis also promotes Th17 differentiation but not the differentiation of Th1, Th2 and Treg [69]. Conversion of glutamine into α-ketoglutarate led to chromatin remodelling that inhibits Th1 differentiation [70]. Glutamine can also be redirected into the gamma aminobutyric acid (GABA) pathway, which shunts the TCA cycle by redirecting glutamine-derived glutamate towards the synthesis of GABA instead of α-ketoglutarate. GABA can be further metabolised into succinic acid for re-entry into the TCA cycle. While GABA is well known for its role in the nervous system, immune cells also express high levels of GABA receptor including monocytes, macrophages, microglial, dendritic cells, T and B cells [71]. GABAergic signalling is typically associated with anti-inflammatory phenotypes reducing T cell proliferation through their decrease of IL-2 production, reduced monocyte, macrophage and DC chemotaxis, phagocytosis and/or cytokine production [71]. On the other hand, sufficient GABA levels are required for the phagocytic and antimicrobial response of macrophages to mycobacterial infection [72].

Other amino acids may also contribute to immune function. Serine is critical to maintain T effector function by supporting *de novo* purine biosynthesis and [73] but also the suppressive activity of Treg by controlling their redox balance through its conversion to glutathione [74]. Absence of glutathione synthase in Treg impairs their function and triggers spontaneous lethal auto-inflammatory disorder. The branched-chain amino acids (BCAA) are catabolised by the enzyme branched-chain amino acid aminotransferase to yield alpha-keto amino acids, a reaction which requires alpha-ketoglutarate, and are further metabolised in the mitochondria by branched-chain alpha-keto acid dehydrogenase. Leucine yields acetoacetate, a ketone body, which can be converted to acetyl-CoA for the acetylation of p300 to activate mTOR [75]. Isoleucine and valine metabolism both yield the TCA intermediate succinyl-CoA.

### 2.8. Mevalonate and Cholesterol Synthesis Pathway Promote Anti-Inflammatory Effects 

The mevalonate pathway is the major pathway for synthesis of cholesterol and isoprenoid lipids including farnesyl diphosphate (FPP) and geranylgeranyl diphosphate (GGPP). This pathway has anti-inflammatory effects but also contribute to the immune response by activating γ𝛿 T cells. Deletion of melanovate kinase has been shown to trigger systemic inflammation by reducing Treg activity [76] as well as the inhibition of HMG-CoA reductase by statins increased Treg in humans [77]. Similarly, the mevalonate pathway was shown to be important for IL-10 production by human Th1 cells [78]. During bacterial infections, APC such as monocytes and DCs upregulate HMG-CoA reductase leading to accumulation of mevalonate metabolites. These metabolites promote human gamma delta T cells (γ𝛿 T cells) effector functions, particularly Vγ9-V𝛿2 T cells [79]. On the opposite, tumour cells were shown to downregulate HMG-CoA reductase as a mechanism to limit the anti-tumour response of γ𝛿 T cells [80]. The mevalonate pathway end-product, cholesterol, is not only the major component of the cellular membrane but can also regulate immune function with its accumulation in macrophages having proinflammatory effects.

Like TCA intermediates, intermediates of the melanovate pathway can also modulate immune cell activation profile through post-translational changes via protein prenylation and geranylgeranylation. Geranylgeranylation has been shown to control toll like receptors (TLR) activation with the absence of protein geranylgeranylation leading to increased TLR and inflammasome activation [76]. Similarly, protein prenylation was shown to limit hyper-production of proinflammatory cytokines as well as constitutive activation of pyrin inflammasome in macrophages [76]. The anti-inflammatory effects of protein prenylation extend to other cell subsets such as B cells by inducing their production anti-inflammatory IL-10 thus promoting anti-inflammatory regulatory B cell differentiation [81].

### 2.9. The Ketogenic Pathway Mediates Anti-Inflammatory Effects

Periods of low glucose availability associated with fasting or caloric restriction, high physical activity, or the consumption of a ketogenic diet, activates ketogenesis in the liver, leading to the production of ketone bodies as a source of energy for the body. β-hydroxybutyrate is the major ketone body produced during ketogenesis and mediates anti-inflammatory effects by blocking nuclear factor kappa B (NF-κB) activation and the assembly and activation of the NLRP3 inflammasome in monocytes and macrophages [82,83]. Administration of β-hydroxybutyrate into mice decreased the severity of inflammasome related inflammatory disease in vivo [83]. Ketone bodies can also elicit immunomodulatory functions by activating metabolite-sensing receptors GPR43 and GPR109 or by inhibiting histone deacetylase activity. Further, ketone bodies may potentially be involved in the metabolic rewiring of immune cells, by inhibiting glycolysis and promoting oxidation pathways. However, while it has been shown that ketone bodies can have immunomodulatory functions, there are currently no reports of ketogenesis occurring in immune cells at this stage.

## 3. Overview of the Gut Immune System

The average adult human is colonised by approximately 3.8×10^13^ bacteria that mostly reside in the gastrointestinal tract, particularly in the colon [84]. The gut immune system has co-evolved to tolerate food antigens as well as these microorganisms, and to concurrently mount an effective immune response towards pathogens. Depending on their localization in the gut, immune cell development and function will be predominantly regulated by dietary components or by the gut microbiota. For example, inducible Treg and macrophages in the small intestine are mostly regulated by dietary components, while in the colon, these subsets are regulated by changes in the gut microbiota linked to diet composition [85]. Both innate and adaptive immune cells are present in the intestine as summarised in the Table 1 and extensively described elsewhere [86]. To limit the interaction with the gut microbiota and to control bacterial expansion, the host has developed both physical and chemical strategies. The first line of defence is ensured by the epithelium, considered as part of the innate immune system. Gut epithelial cells involved in host defence include enterocytes, Paneth cells and goblet cells, and also Microfold cells, which are present only in the small intestine. Enterocytes ensure a physical barrier through their expression of tight junction proteins, limiting the translocation of bacteria and their endotoxins. Tight junction proteins include occludin, claudin and zonula occludens that form the core of the tight junction structure, while claudin regulates the paracellular integrity. Goblet cells produce mucus that also act as a physical barrier while Paneth cells produce anti-microbial peptides to regulate intestinal bacterial load. Disruption to the equilibrium between gut bacteria and the host, observed in germ-free mice deficient in bacteria or in antibiotic-triggered dysbiosis, have major consequences on the gut immune system (Table 1). This equilibrium is bi-directional with inflammatory diseases also affecting the gut microbiota as described elsewhere [87,88]. Finally, like for immune cells in other compartment, the metabolic activity of gut immune cells has to be adapted to their energy demand (summarized in Table 1). While the field of immunometabolism is expanding, there are still a lot of unknown around the metabolic profile of gut immune cells as most studies have used splenic or bone marrow-derived cells as models due to the technical limitations.

## 4. Nutrient Sensing and Gut Immunometabolism

### 4.1. Dietary Protein and Gut Immunity

The spatiotemporal impact of proteins, as well as their qualitative and quantitative aspects, affect both the maturation and function of the gut immune system. The exposure to dietary proteins at weaning is necessary for small intestine immune development with mice fed on protein-deficient diet exhibiting underdeveloped gut-associated lymphoid tissues, lower number of intraepithelial lymphocytes, reduction of Peyer’s patches and decreased secretory immunoglobulin A (IgA) levels [135]. It is interesting to note that fasting drastically decreased Peyer’s patches B cells and antigen-specific IgA response. This was linked to the redirection of gut B cells back into the bone marrow that migrated to the gut under feeding conditions [136]. Whether dietary protein affects the mobilisation of bone marrow B cells to the gut is unknown. Proteins are also the major source of antigens necessary for the development of immune tolerance towards food particularly by inducing Treg in the small intestine at weaning [85]. While the gut microbiota induces Treg in the colon [137], the effect of dietary protein is specific to the small intestine. The mechanism behind this observation is not fully elucidated but a protein-deficient diet with defined amino acids does not induce Tregs in the small intestine [85], suggesting the generation of antigen-specific Treg rather than a metabolic effect involving mTOR activation. The route of entry of protein is also critical with parenteral nutrition but not enteral nutrition decreasing the proportion of small intestine IL-10 producing macrophages [138] and intraepithelial lymphocytes [139]. Like small intestine inducible Treg, diet-derived protein maintained the IL-10 producing macrophage population independently of monocyte recruitment and of the gut microbiota. Rapamycin treatment mimicked these effects suggesting that the activation of mTOR by dietary amino acids is necessary to maintain IL-10 production in macrophages [138]. Specific deletion of *Raptor* in dendritic cells confirmed the key role of mTOR for IL-10 production with decreased production in these deficient colonic dendritic cells [140]. In this model, CD11c^+^CD11b^+^F4/80^+^ dendritic cells produced less IL-10 and mice were more susceptible to colitis. However, multiple pathways might regulate gut immune-derived IL-10 as in vitro stimulation of macrophages with the amino acid leucine, a potent mTOR activator, could not fully rescue the production of IL-10 [138]. Glutamine metabolism might be another important mechanism as parenteral nutrition supplemented with glutamine could rescue IL-10 production of intraepithelial lymphocytes and restore epithelial integrity [139]. The fact that glutamine utilisation through glutaminolysis skew macrophage polarization towards M2 [141] suggests that glutamine metabolism might be important to maintain small intestine IL-10 producing macrophages and potentially IL-10 production from other subsets.

The amount and type of proteins can also affect gut homeostasis. Diet containing animal-derived protein [142] or a high protein content in diet (53%) have been shown to aggravate DSS induced colitis while moderately high protein content (30%) accelerated gut healing after DSS challenge compare to a low protein diet (15%) [143]. These mechanisms seem to involve changes to gut microbiota composition rather than a direct effect on the immune system.

### 4.2. Dietary Lipids and Gut Immunity

Dietary medium- and long-chain fatty acids have been shown to boost Th1 and Th17 differentiation in the small intestine, aggravating the development of experimental autoimmune encephalomyelitis [144]. This effect involves the direct activation of mitogen-activated protein kinase (MAPK) pathways by fatty acids, independent of GPCR or nuclear receptor activation [144]. While this study focused on the medium-chain fatty acid lauric acid, how other fatty acids, particularly unsaturated versus saturated fatty acids, might affect gut immunity remains unknown. The gene that most dramatically decreased in the presence of lauric acid was *c-Maf* [144], which is highly expressed in retinoic acid-related orphan receptor gamma t (RORγt)^+^ Treg in the small intestine, colon and Peyer’s patches. Deletion of *c-Maf* in Treg decreased their expression of IL-10 while increased their production of IL-17 and interferon gamma (IFN-γ) [145]. Absence of *c-Maf* was also correlated with gene enrichment of the phosphoinositide 3-kinases (PI3K)/ Protein kinase B (AKT)/mTOR pathway as well as of the glycolytic protein MYC targets, suggesting that lauric acid might bias the Treg:Th17 balance towards Th17 in the small intestine by promoting glycolysis secondary to *c-Maf* downregulation. Long-chain fatty acids can also modulate intestinal B cell activity with increased small intestine and fecal IgA levels in mice fed on isocaloric diet enriched in palmitic acid under basal and oral antigen challenge conditions [146]. Palmitic acid promoted plasma cell IgA production through its intracellular conversion into sphingolipid, independently of TLR4 activation. Innate immune cells are also affected by dietary fat as activation of peroxisome proliferator-activated receptor gamma (PPARγ) by conjugated linoleic acid protected mice from DSS induced colitis [147] and the deletion of PPARγ in macrophages aggravated dextran sulfate sodium (DSS)-induced colitis [148]. Activation of PPARγ in macrophages modulate their basal respiration and elicit anti-inflammatory effects by promoting M2 polarisation via increased glutamine metabolism [149]. Whether PPARγ activation by dietary fat could maintain macrophage anti-inflammatory phenotype in the gut is unknown.

While a hypercaloric high-fat diet is typically associated with a pro-inflammatory immune phenotype, it is not associated with increased immunity against infections. Animals fed on a high-fat diet had impaired response to salmonella [150] and *Listeria monocytogenes* [151]. One potential explanation is that hypercaloric high-fat diet induced hypercholesterolemia, which has been shown to inhibit the PPP, reduce LPS-induced pro-inflammatory cytokine production in macrophages [21], and thus their capacity to get activated during infection.

### 4.3. Dietary Carbohydrates and Gut Immunity

Like proteins and lipids, carbohydrates also affect gut immunity. Supplementation of drinking water with glucose exacerbated the development of autoimmune colitis in model of CD4^+^CD25^-^CD45RB^hi^ T cell transfer into *Rag1*^-/-^ mice associated with increased colonic Th17 cells [152]. The mechanism identified was that high glucose condition increased ROS generation, which in the presence of IL-6 and TGF-b promoted Th17 differentiation. Interestingly, the presence of glucose did not affect T cell metabolism as T cells isolated from mice treated with glucose had similar oxygen consumption rate and glycolytic profile as T cells from control mice. It is important to note that the metabolic profile of intestinal T cells has not been investigated, thus the impact of high glucose exposure on their metabolic profile cannot be excluded. Also, whether ROS activation in the gut might increase Th17 differentiation is unknown. In line with this, short-term intake of a diet enriched in simple carbohydrate was associated with increased colonic neutrophil infiltration, enhanced levels of IL-6, IL-1β and TNF and elevated gut permeability [153].

The beneficial anti-inflammatory effects of a ketogenic diet may thus be explained by its low-carbohydrate content; however, it is hard to differentiate whether its immune effects are due to the high intake of fat, the low intake of carbohydrate, or both. This diet promotes Treg while decreases Th17 although these effects were attributed to changes within the gut microbiota [154] and a direct impact on immune cells has to be confirmed.

### 4.4. Dietary AhR Ligands and AhR Activation

Aryl hydrocarbon receptors (AhR) are nuclear receptors that upregulate the expression of xenobiotic-metabolising enzymes like the cytochrome P450, family 1, subfamily A, polypeptide 1 (CYP1A1) enzyme when activated. A wide range of ligands exists for this receptor, such as indole-3-carbinol (I3C) and indole-3-acetonitrile (I3AC) from the digestion of cruciferous vegetables, or ligands derived from bacterial metabolism of tryptophan, covered in the next section.

AhR is expressed by colonocytes, CD4^+^ T cells, gd T cells, antigen-presenting cells and innate lymphoid cells. Activation of AhR promotes the expression of IL-22 and IL-17 by γ𝛿 T cells and ILC3 but also the expression of IL-10 and IL-21 by Type 1 forkhead box P3 (FoxP3)^-^ Treg cells [155]. Interestingly the effect of AhR activation can be ligand-specific, with 2,3,7,8-Tetrachlorodibenzo-p-dioxin (TCDD) promoting Treg, while 6-formylindolo[3,2-b]carbazole (FICZ) promoting Th17 differentiation. While AhR mostly mediates its effect by altering gene expression, its activation with the natural agonist norisoboldine has been shown to promote colonic Treg by inhibiting glycolysis under hypoxic conditions [156]. This effect was linked to the decreased expression of glucose transporter-1 (GLUT1) and the enzyme hexokinase 2 in CD4^+^ T cells, which favoured their differentiation to Treg [157]. Whether different AhR ligands differentially affect immune cell metabolic activity remains to be determined. Finally, how diet composition might interfere with the immune impact of AhR ligands is unknown. A recent study has shown that the AhR ligand TCDD reduced tumor growth in mice fed on an Omega-3 fatty acid-enriched diet while an Omega-6 fatty acid-enriched diet aggravated tumor growth [158]. While this study focused on the direct impact of AhR on tumor cells, the differential effect of AhR ligands on the Treg/Th17 balance depending on the dietary context cannot be excluded.

### 4.5. Vitamins

In the gut, naïve B cells located in the Peyer’s patches and IgA producing plasma cells present in the lamina propria have different energetic needs. Naïve B cells rely on the TCA cycle activity exclusively while IgA producing plasma cells rely on glycolysis coupled with the TCA cycle. This feature was specific to intestinal IgA^+^ B cells as splenic plasma cell did not show increased glycolytic activity under basal condition. Dietary vitamin B1 is an essential cofactor required for the TCA cycle enzymes pyruvate dehydrogenase and α-ketoglutarate dehydrogenase. Indeed, depletion of dietary vitamin B1 had dramatic consequences on gut lymphoid structures, evident by the reduction of Peyer’s patches number, by the size of B cell follicles and lower number of naïve B cells, while IgA^+^ plasma cells were not affected. This reduced number of naïve B cells was linked to a defect in B cell lymphopoiesis, which was reversible once Vitamin B1 was reintroduced in the diet. The highest dependency on Vitamin B1 in naïve cells was also associated with their highest expression of the thiamine transporter 1 compare to IgA^+^ plasma cells [89].

Vitamins also affect other immune subsets particularly Treg with dietary vitamin A, D, B9 (folic acid) and B3 (niacin) promoting colonic Treg development. Folic acid had been shown to have a direct impact on colonic Treg maintenance with its absence impairing colonic Treg exclusively and exacerbating the susceptibility of mice to TNBS-induced colitis. This effect of folic acid involved the binding of folic acid on the folic receptor 4 expressed by Treg which modulated *Bcl2* expression and thus controlled Treg apoptosis [159].

Vitamin D and Vitamin A also induce gut Treg by acting on tolerogenic DC. Gut tolerogenic CD103^+^ DCs induce Treg in the mesenteric lymph node and Peyer’s patches by converting vitamin A-derived retinol into retinoic acid through the activity of the enzyme retinaldehyde dehydrogenase (RALDH). Retinoic acid also supports the generation of IgA-producing plasma cells as well as induce the gut-homing receptors C-C chemokine receptor type 9 (CCR9) and α4β7. Retinoic acid regulates gene expression, such as FoxP3 characteristic of Treg, through its interaction with the nuclear receptors retinoic acid receptor (RAR) and retinoic X receptor (RXR) that specifically binds retinoic acid responsive element [160]. FoxP3 has been shown to regulate Treg metabolic activity by inducing mitochondrial genes, which increased electron transport chain protein capacity as well as mitochondrial respiration [161]. These findings were on splenic inducible Treg and have to be confirmed on gut inducible Treg.

Vitamin D is also important for gut Treg as a diet deficient in vitamin D decreased the proportion of inducible RORγt^+^ colonic Treg in mice [162]. It has been shown that the active form of Vitamin D3, 1,25-dihydrovitamin D3, could promote the conversion of human monocytes into tolerogenic dendritic by maintaining aerobic glycolysis via activation of PI3K/AKT/mTOR pathway [113]. Whether these metabolic changes apply to CD103^+^ tolerogenic DC in the gut remains elusive. 1,25-dihydrovitamin D3 can also directly promote Treg differentiation through the induction of FoxP3 gene expression. The induction of gut-inducible Treg was also shown to be mediated by changes to the gut microbiota, associated with the decreased abundance of gut bacteria species that produces short-chain fatty acids [162], important Treg inducer as developed in the next section.

### 4.6. Impact of Food Additive on Gut Immunity and Immunometabolism

Food additives encompass a range of substances which alter the colour, flavour, texture and increase the shelf life of food. In recent years, additives have become ubiquitous in processed foods and are categorised as GRAS (generally regarded as safe). However, studies into the effects of chronic exposure to food additives have been mostly neglected until recently. Mounting evidence suggests that additives have a profound impact on the immune system and contribute to chronic diseases such as obesity [163], metabolic syndrome [164], type 2 diabetes [165] and inflammatory bowel disease (IBD) [164]. While mechanisms behind these effects are still under investigation, it is becoming increasingly clear that these are in part due to metabolic programming of immune cells. The number of food additives approved by the U.S Food and Drug administration is currently more than 3000 and includes things as citrate and salt, which can be broadly classified into “natural”, as well as food colourants such as titanium dioxide, which fall under the “artificial” category. While such labels may influence the consumer, this classification has little to no value in predicting outcomes on immune health, with both natural and artificial additive having varied effects on gut homeostasis (Table 2).

For instance, citrate, a naturally occurring metabolite involved in carbohydrate and lipid metabolism is widely used as a food additive. Exogenous citrate enters cells via the citrate transporter and is rapidly metabolised into acetyl-CoA in the cytoplasm [166]. In both DC and macrophages, cytosolic citrate is a substrate for protein acetylation, histone acetylation and fatty acid synthesis. Citrate can also be metabolised into pyruvate, through reactions that in turn yield NADPH, increasing NO as well as ROS production. While NO inhibits OXPHOS, ROS stabilises HIF-1α and shift macrophage metabolism towards glycolysis and thus potentially towards M1 [167]. Citrate accumulation in macrophages is thus associated with increased inflammatory response also observed when citrate was added to LPS treatment in THP-1 monocytes [168]. Mice fed on citrate in combination with a high sucrose diet had higher levels of IL-1b, TNF-a and IL-6 in the adipose tissue than mice fed high sucrose alone [166] suggesting an impact of citrate on M1 differentiation in vivo. However, whether citrate affects gut macrophage activation as well as gut inflammation is unknown.

Salt similarly has vast impacts on immune function. While sodium chloride is used as a preservative and flavour enhancer in a wide range of food products, high intake of salt is harmful to health, exacerbating risks of numerous diseases particularly cardiovascular diseases. High salt diet has been shown to spontaneously trigger colonic inflammation linked to higher levels of colonic IL-23, of recruitment of neutrophils [169] as well as of activated T cells, macrophages and DCs [170]. Excess salt consumption has been shown to promote a pro-inflammatory phenotype by increasing Th17 and M1 macrophages while reducing Treg and M2 macrophages, leading to exacerbated autoimmune disease, inflammatory bowel disease and impaired wound healing [171,172,173,174].

Salt induced inhibition of M2 macrophages was associated with decreased maximal respiratory capacity, decreased glycolysis and mTOR activation while promoted glycolysis in resting macrophages [171]. Consumption of high salt diet has been shown to upregulate the expression of the enzyme serine/threonine kinase 1 (SGK1), which maintains sodium homeostasis. High SGK1 expression has been observed in colon of mice fed on high salt diet [169] as well as in Th17 and Treg [175]. SGK1 has been shown to support glucose uptake and glycolysis at the cellular level [176]. High salt diet can also change the balance Th17/Treg indirectly by depleting bacteria of the genus *Lactobacillus* [177], which via their release of indole metabolites decreased the differentiation of Th17.

Another major food additive is titanium dioxide (TiO_2_) which affects gut microbiota composition, increases colonic inflammation [178] as well as induces preneoplastic lesions [179]. Macrophages phagocyte these nanoparticles and trap them in multivesicular bodies. TiO_2_ uptake affect macrophage mitochondrial activity by decreasing the TCA cycle and ATP production and promote ROS production, which in turn activate their inflammatory response [180]. Whether TiO_2_ switches macrophage metabolic profile towards glycolysis is unknown as well as its impact on the metabolism of other immune cell such as CD8+ cells activated under TiO_2_ treatment [178]. The impact of food additives on immune cells is summarised in Table 2 and an overview of the impact of dietary components on immunometabolism is presented in Figure 1.

High-protein diet can regulate immunometabolism by providing a source of glutamine to drive glutaminolysis and IL-10 production in M2 macrophages. Branched-chain amino acids (BCAA) from protein also activates mechanistic target of rapamycin (mTOR) to also promote IL-10 production in macrophages, and also in dendritic cells (not shown). The dietary lipid acid lauric acid promotes Th1/Th17 differentiation by downregulating c-Maf and upregulating serine/threonine-protein kinase (SGK-1), associated with increased mTOR activation and glycolysis. Palmitic acid induces immunoglobulin (IgA) production in B cells by promoting *de novo* sphingolipid synthesis while the conjugated linoleic acid drive M2 macrophage polarization by activating peroxisome proliferator-activated receptor gamma (PPARγ) to increase glutaminolysis. The food additive salt also promotes Th17 cell differentiation, potentially by also increasing SGK-1 expression. Along with citrate and titanium dioxide (TiO_2_), salt promotes pro-inflammatory M1 macrophage differentiation by enhancing glycolysis. Micronutrients such as Vitamin D can drive the differentiation of tolerogenic dendritic cells by promoting mTOR activation and aerobic glycolysis. Vitamin A is also metabolised by the enzyme retinaldehyde dehydrogenase (RALDH) expressed by gut tolerogenic dendritic cell to produce retinoic acid, which promotes regulatory T cell (Treg) differentiation. Similarly, dietary-derived aryl hydrocarbon receptor (AhR) ligands such as norisoboldine can directly promote Treg differentiation by inhibiting glucose transporter-1 (GLUT1) as well as glycolysis enzyme hexokinase 2 (HK2).

## 5. Dietary-Induced Bacterial Metabolites and Gut Immunometabolism

### 5.1. Microbial Metabolism of Complex Carbohydrate Produces Short-Chain Fatty Acid with Anti-Inflammatory Properties

Complex carbohydrates or microbiota-accessible carbohydrates are found in dietary fibre and resistant starch, and are a major source of energy for colonic bacteria [137]. Gut bacteria can extract energy from complex carbohydrates through the process of fermentation, releasing short-chain fatty acids (SCFA) as by-products [54]. They also have prebiotic properties by favouring the growth of beneficial bacteria that can utilise these substrates.

SCFA are volatile fatty acids containing six or less carbon with acetate (C2), propionate (C3) and butyrate (C4) among the most abundant metabolites produced by the gut microbiota. SCFA modulate immune cell differentiation or activity either through epigenetic changes by inhibiting histone deacetylase (HDAC) activity, or through the activation of specific GPCR expressed on immune cells such as GPR41, GPR43 and GPR109 [54]. SCFA may also influence host intracellular metabolism, by serving as substrates for ketone bodies (butyrate) or acetyl-CoA/citrate (acetate) productions, or for post-translational acetylation of metabolic enzymes such as GAPDH (acetate) to activate glycolysis [193]. While the activation of GAPDH by acetylation was observed when acetate was acutely released from the liver during infection, this phenomenon has not been reported with gut-derived acetate.

The first cells in contact with SCFA are enterocytes, and butyrate is the primary source of energy for colonocytes. Butyrate undergoes beta-oxidation that in turn raises ATP levels via oxidative phosphorylation. This process inhibits autophagia that is significantly increased in germ-free mice devoid of luminal butyrate [194]. The utilization of oxygen during OXPHOS is critical to maintain an anaerobic gut environment as depletion of butyrate-producing bacteria by antibiotic treatment increased the diffusion of oxygen in the gut lumen favouring the growth of facultative anaerobe pathobionts [195]. Thus, a high fibre diet could directly promote a healthy gut microbiota by selecting for bacteria with the enzymes necessary for its fermentation, and indirectly by reducing oxygen levels in the gut through the metabolism of butyrate. We have also shown that SCFA maintained gut epithelial homeostasis through NLRP3 inflammasome activation via GPR43 and GPR109a activation [196]. Whether this effect is associated with changes in enterocyte metabolic profile such as beta oxidation and OXPHOS is unknown.

Innate lymphoid cells (ILC) are innate immune cells with lymphoid features that resembles T lymphocytes. ILC3 are important mediators of intestinal homeostasis through their production of IL-22 which maintains epithelial homeostasis [197] and to limit inflammation [198]. The highest abundance of ILCs is in the intestinal lamina propria where they are key communicators between the host and the gut microbiota. ILC expresses all major SCFA receptors and are thus sensitive to regulation by dietary intake of fibre. The activation of GPCR has dual effects on ILC development with GPR43 activation directly promoting ILC3 proliferation [199] while GPR109a activation decreasing ILC3 proportion in the colon [200] and in terminal ileal Peyer’s patches [201]. SCFA activation of GPCR also regulate cytokine production from ILCs and T cells with butyrate activation of GPR41 increasing their production of IL-22. This effect was also due to butyrate-mediated HDAC inhibition as well as the upregulation and activation of both AhR and HIF-1α [202]. This change in HIF-1α is likely due to butyrate mediated mTOR activation in these cells. HIF-1α is known to promote glycolysis, however whether the secretion of IL-22 in ILC and CD4^+^ T cells is linked to a shift towards glycolysis has not been tested. On the opposite, butyrate has been shown to inhibit glycolysis in gut macrophages [203] suggesting finely tuned immune-metabolic activation by SCFA. 

SCFA also regulate the activity of intestinal macrophages and DC. Butyrate-mediated HDAC inhibition drove the differentiation of intestinal macrophage from blood derived monocytes and enhanced their antimicrobial activity independently of GPCR signalling [203]. Contrary to its effect in ILC3, butyrate decreased mTOR activity and inhibited glycolysis, which was compensated by higher mitochondrial energy metabolism. This enhanced macrophage antimicrobial activity was confirmed by oral administration of butyrate limiting the systemic dissemination of *Salmonella* and *C. rodentium* during oral infection. The effect of butyrate on macrophages was through an imprinting effect as exposure to butyrate during their differentiation rather than during infection was necessary for their enhanced microbicidal activity [203]. Butyrate also plays dual roles on macrophage, and could favour colonic M2 macrophage differentiation in mice treated with antibiotics and receiving water supplemented with butyrate [204]. While this study shows that butyrate increases OXPHOS and lipid metabolism in vitro, this was shown in bone marrow-derived macrophages and thus whether butyrate has these effects in colonic macrophages is unknown.

On the other hand, both acetate and butyrate impacted on gut DC by enhancing the tolerogenic function of mucosal CD103^+^ DC by enhancing RALDH enzyme activity [205].

This effect was through GPR43 and GPR109 signalling and results in increased antigen-specific Treg with elevated expression of the gut homing receptor CCR9. As a result, high fibre feeding improved oral tolerance and decreased the severity of food allergy [205]. The role of mTOR in the induction of CD103^+^ DC was recently shown in the lung with the decreased proportion of CD103^+^ DCs in mice with specific deletion of mTOR in DC, leading to exacerbated allergic reaction [206]. Whether SCFA also affect CD103^+^ DC homeostasis via mTOR modulation or more broadly via changes in their metabolic activity remains unknown.

SCFA also affect the metabolic profile of intestinal B cells, through their conversion into acetyl-CoA fuelling the TCA cycle coupled with increased OXPHOS as well as fatty acid synthesis [207]. This raise of energy production might explain the activation of mTOR in these cells, which promotes glycolysis to provide the energy necessary for plasma cell differentiation and their production of IgA and immunoglobulin G (IgG) [208].

On the contrary, inhibition of HDAC by the SCFA butyrate and propionate was reported in splenic B cells to impair class-switch DNA recombination, reducing the expression of *Aicda* and *Prdm1*, as well as their differentiation to plasma cell. As a result, SCFA treatment reduced serum levels of antigen-specific and total IgG and IgA and protected mice from lupus, an autoantibody mediated disease [207]. This result suggests that SCFA differentially regulate antibody production under homeostatic versus pathogenic condition and through different mechanisms.

SCFA also significantly impact the T cell compartment in the gut. mTOR activation by SCFA (acetate and propionate) in CD4^+^ T cells contribute to the development of both Treg, Th1 and Th17 cells depending on the cytokine environment. This impact of SCFA on mTOR activation has been confirmed with the SCFA pentanoate, which on the opposite inhibited Th17 development [209]. SCFA induced Th1, Treg and Th17 through HDAC inhibition and was important to maintain gut homeostasis as acetate treatment protected mice from colitis induced by *C. rodentium* or by anti-CD3 stimulation [210]. While this study shows that neither GPR41 nor GPR43 are behind the effects of SCFA on T cell differentiation, another study has shown that SCFA, particularly propionate, induced colonic Treg in a GPR41 dependent manner [211].

SCFA also promote memory CD8+ T cell with acetate increasing glycolysis through the acetylation of GAPDH [193] and butyrate favouring glutaminolysis and fatty acid oxidation in a GPR43 and GPR41 dependent manner [212]. However, these studies were focused on splenic CD8^+^ T cells and thus the effect of SCFA on gut memory CD8^+^ T cells remain unknown. However, if validated the effect of SCFA on gut CD8^+^ memory T cells would more likely be through fatty acid oxidation and glutaminolysis rather than glycolysis, which was observed only when acetate was acutely released from the liver during infection [193].

Other metabolites such as succinate are produced as intermediates from the metabolism of complex carbohydrates [213]. High-fiber diet has been shown to elevate microbial succinate production [214] and is used by the intestinal epithelium as a substrate for gluconeogenesis. Increased intestinal gluconeogenesis via succinate was shown to improve glucose homeostasis [215] and may account for the beneficial effects of dietary fiber. On the other hand, increased intestinal concentration of succinate is reported in patients with inflammatory bowel disease [216], which is consistent with the pro-inflammatory role of succinate on macrophages [167]. While the impact of succinate on macrophage metabolism has extensively been studied, whether microbiota-derived succinate can have the same impact on immune cells is unknown.

### 5.2. Microbial Metabolism of Amino Acid Produces Metabolites with Immunomodulatory Properties

Aromatic amino acids can be metabolised by the gut microbiota to produce various tryptophan-, phenylalanine- and tyrosine-derived immunomodulatory metabolites such as indole-3-pyruvic acid, indole-3-acetic acid (IAA) and indole-3-propionic acid, reviewed extensively elsewhere [217]. Many of these metabolites are produced exclusively by the gut microbiota and can be detected in the periphery blood in humans [218].

Tryptophan-derived metabolites such as indolelactic acid (ILA) and indoleacetic acid (IAA) regulate gut homeostasis by promoting IL-22 production via AhR activation. Maternal intake of a high fat diet was shown to alter offspring immunity by increasing IL-17-producing ILC3 in the gut, which increased susceptibility to gut injury following exposure to LPS and platelet-activating factor [219]. The specific mechanism is likely through the expansion of *Firmicutes* within the gut microbiota, specifically by increasing *Lactobacilli* capable of utilising tryptophan to produce AhR ligands to induce ILC3 [219]. Similarly, ILA produced by the metabolism of dietary tryptophan by *Lactobacillus reuteri* induces the differentiation of gut intraepithelial CD4^+^CD8αα^+^ T cells via AhR activation [220]. Tryptophan-derived metabolites, such as kynurenic acid, are ligands for the GPCR GPR35. GPR35 is expressed by inflammatory CX3CR1^+^ macrophages in a microbiota-dependent manner and is upregulated during colonic inflammation [221]. Its activation by lysophosphatidic acid was shown to improve inflammation in a mouse model of colitis [221]. Butyrate has been reported to decrease IDO1 expression in epithelial cells [222], redirecting tryptophan metabolism towards the serotonin pathway that produces the AhR ligand 5-Hydroindole-3-acetic acid, which raised splenic regulatory B cells [223]. Whether colonic regulatory B cells or their metabolic profile are regulated by butyrate in this manner remain unknown. In host, a low amount of tryptophan in diet is preferentially converted into kynurenine by the enzyme IDO1. High tryptophan diet decreased gut permeability and decreased the inflammation in a model of gluten immunopathology in mice [224]. On the opposite, patients with celiac diseases had lower tryptophan and AhR agonists in stool while kynurenine metabolites were increased compare to healthy participants [224]. These observations highlight that the metabolism of tryptophan by both the host and the microbiota is important for gut homeostasis.

### 5.3. Impact of Secondary Bile Acids on Gut Immune Profile and Immuno-Metabolism 

Bile acids are synthesized in the liver from cholesterol and assist in the absorption and digestion of lipids and fat-soluble vitamins [225]. Hepatocytes produce the primary bile acids, cholic acid and chenodeoxycholic acid, which are excreted in the small intestine, reabsorbed and recycled via the enterohepatic circulation. Less than 5% escape reabsorption to reach the colon where they are modified by the colonic bacteria into secondary bile acids [225]. Bile acids can interact with host cells either via GPCRs such as the G protein-coupled bile acid receptor GPBAR1 also known as TGR5 or via the nuclear receptors farnesoid X receptor (FXR) and vitamin D receptor. Extensive review on bile acid receptors have been reported elsewhere [226]. TGR5 is highly expressed by the intestinal epithelium, as well as in immune cells, particularly monocytes and macrophages [227]. In the gut, TGR5 activation promotes the regeneration of the intestinal epithelium by increasing intestinal stem cells activity [228]. This was shown to be important for the maintenance and recovery of the intestinal barrier during colitis. TGR5 activation reduced TNF production in in vitro differentiated macrophages resembling human intestinal lamina propria CD14^+^ macrophages that mediate the pathology of Crohn’s disease [227]. Deletion of TGR5 reduced macrophage chemotaxis and inflammatory response to LPS in mice fed on high fat diet via impaired AKT-mTORC1 activation [229], but it is not known whether this impact on mTOR affects their metabolic profile. Activation of FXR also has anti-inflammatory effects, associated with decreased macrophage activation and preserved epithelial integrity in mouse models of colitis [230]. The impact of conjugated bile acids also extends to the adaptive immune system with the activation of vitamin D receptor by a mixture of 3-oxo lithocholic acid (LCA), LCA and deoxycholic acids promoting the differentiation of colonic naïve T cells towards RORγt^+^ inducible Treg [231]. This effect required microbial modification of bile acids with mice fed on a minimal diet having less colonic RORγt^+^ inducible Treg [231]. IsoalloLCA is another secondary bile acid shown to promote Treg differentiation by increasing mitochondrial respiration and ROS production, which led to upregulation of the FoxP3 gene [232]. Interestingly, IsoalloLCA also inhibited IL-17 expression in Th17 cells [232]. Whether the inhibition of IL-17 was linked to changes in T cell metabolism has not been described. An overview of the impact of bacterial-derived metabolites on immunometabolism is presented in Figure 2.

## 6. Conclusions

The intestinal immune system must adapt to a broad range of challenges to maintain homeostasis and health. It must maintain tolerance towards food antigens as well as gut microbiota-derived products while mounting an effective immune response towards pathogens. The induction of Treg by gut bacteria and food antigens forms part of this tolerance strategy, which not only maintains gut homeostasis but also decreases susceptibility to inflammatory diseases that can develop distally such as in the central nervous system [144]. The intestinal immune system is therefore critical for overall health and requires energy to fuel its processes, which includes cell division, differentiation and effector activity. As emphasized throughout this review, the status and phenotype of an immune cell is linked to different metabolic pathways depending on its specific energetic and anabolic requirements, which may be regulated by substrate availability. Both dietary components and bacterial metabolites are substrates that can activate different metabolic pathways to influence the differentiation and activity of gut immune cells affecting gut homeostasis and overall health.

However, we can only speculate on how these substrates influence gut immunometabolism as most studies focus on peripheral rather than intestinal cells. Studies specific to gut immune cell metabolism are limited due to the technical challenges associated with the isolation of sufficient number of cells and with the in vitro culture conditions far from the native gut environment. New flow cytometry tools such as SCENITH [233] could bridge this gap to revolutionise the field of immunometabolism and more broadly of medicine. Indeed, the fact that specific food components or bacterial metabolites could favour specific metabolic pathway to drive the activation status of immune cells activation status has great therapeutic potential. This could lead to personalised nutritional strategies to rewire gut immunometabolism to prevent or treat diseases. Similarly, probiotic and postbiotic interventions could be used for this purpose. One further consideration is that the efficacy of specific intervention can be influenced by the individual’s nutritional context and their gut microbiota composition. The geometric framework for nutrition [234] is one statistical tool that can be used to unravel this complexity. Finally, a better understanding on why particular diets are detrimental could lead to stricter regulation, particularly in the area of food additives, which is highly prevalent in processed food with potential detrimental impact on health.

## Figures and Tables

**Figure 1 nutrients-13-00823-f001:**
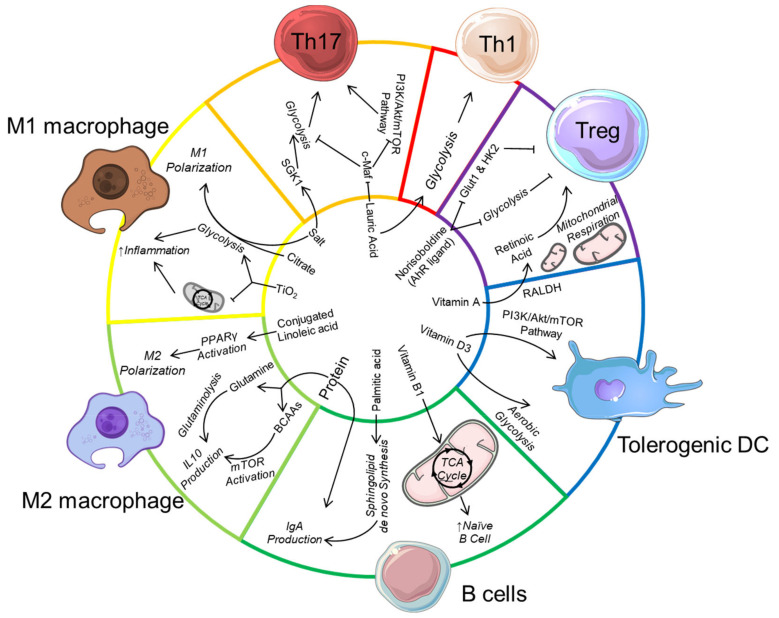
Impact of dietary-derived metabolites on host immunometabolism and gut immune cell function.

**Figure 2 nutrients-13-00823-f002:**
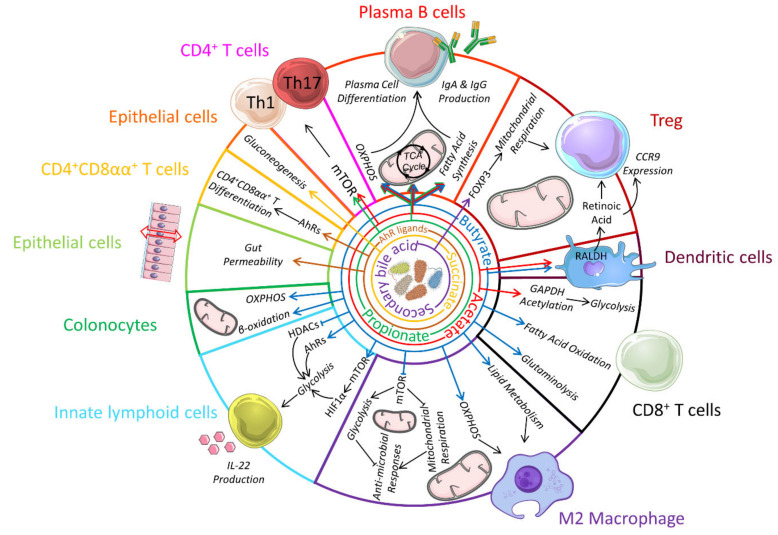
Impact of bacterial-derived metabolites on host immunometabolism and gut immune cell function. Butyrate can be oxidised by colonocytes to generate energy via oxidative phosphorylation (OXPHOS), or to promote metabolic rewiring of immune cells such as M2 macrophages and CD8^+^ memory T cells towards fatty acid oxidation and OXPHOS. Butyrate also inhibits histone deacetylase (HDAC), and upregulate aryl hydrocarbon (AhR) receptor and hypoxia-inducible factor 1-alpha (HIF-1α) expression to promote glycolysis in innate immune cells to drive their production of IL-22. Acetate and propionate induces Th1/Th17 differentiation by activating the mechanistic target of rapamycin (mTOR), while acetate and butyrate promote regulatory T cell (Treg) differentiation by enhancing retinaldehyde dehydrogenase (RALDH) enzyme activity in tolerogenic dendritic cells to produce retinoic acid. Retinoic acid also induce the gut homing receptor C-C chemokine receptor type 9 (CCR9) in Treg. Acetate also enhances memory CD8^+^ T cell effector function by enhancing glyceraldehyde 3-phosphate dehydrogenase (GAPDH) enzyme activity to drive glycolysis. Butyrate, acetate and propionate can promote plasma B cell differentiation and antibody production by activating OXPHOS and fatty acid synthesis, and to promote immunoglobulin A (IgA) and immunoglobulin G (IgG) production. In the gut, succinate act as a substrate for gluconeogenesis in epithelial cells, and may also potentially serve as a pro-inflammatory molecule (not shown). Secondary bile acids, such as 3-oxo lithocholic acid, can directly induce the differentiation RORγt Treg in the colon while AhR ligands can decrease gut permeability, as well as drive the differentiation of CD4^+^CD8αα^+^ T cells.

**Table 1 nutrients-13-00823-t001:** Cells of the gut immune system, their main anatomical location in the gut, their function and their regulation by diet and the gut microbiota.

Cell Type/ Markers	Anatomical Location in the Gut	Functions	Metabolic Profile	Dietary Impact	Microbial Impact		Reference
					Antibiotic Impact	Germ-Free Conditions	
**B cells**	Lamina propria, Peyer’s patches, mesenteric lymph node	Immunoglobulin production (IgA, IgG, IgM) Host-microbiota mutualism	Intestinal naïve B cell utilizes TCA-OXPHOS IgA^+^ plasma cell utilizes glycolysis-TCA-OXPHOS	High fibre diet increases plasma cells ↑ IgA after weaning ↑ Germinal centre B cell development after weaning ↑ IgA^+^ plasma cell after weaning	↓ in SI, colon and Peyer’s patches; similar in mesenteric lymph node	↓ IgA and IgG production in SI	[89,90]
**Regulatory B cell**	Lamina propria, mesenteric lymph node	Intestinal inflammation suppression through IL-10 production and repression of IL-1 and STAT3-related inflammatory cascades	Little is known about the development and metabolism of Breg cells but they can increase glycolysis during activation				[91]
**CD4^+^ T cell**	Lamina propria, Peyer’s patches, mesenteric lymph node	*Th1:* anti-viral and -bacterial responses *Th2:* anti-parasite responses *Th17:* anti-fungi and anti-bacterial responses	Glycolysis is dominant during activation of Th cells TCA cycle/OXPHOS is dominant for long-lived and naïve CD4^+^ T cells	Th2-bias responses and lack of Th1 responses before weaning TCR repertoire is polyclonal before weaning but become restricted/oligoclonal after weaning	↓ in Peyer’s patch, and SI; similar or ↓ in mesenteric lymph node; ↓ memory CD4^+^ T cell in SI, colon and mesenteric lymph node	↓ in SI, and mesenteric lymph node; ↓ or similar in colon	[92,93,94]
**CD8^+^ T cell**	Lamina propria, Peyer’s patches, mesenteric lymph node	Cytotoxicity Mucosal defence	Naive: FAO and OXPHOS Activated: Glycolysis and OXPHOS	↑ in intestine recruitment after weaning (mostly CD8αβ^+^ TCR^+^)	↓ in SI, and colon, ↑ or similar in mesenteric lymph node and Peyer’s patch Similar IFNγ production in SI	↓ in mesenteric lymph node and similar in SI ↓ IFNγ production in SI, colon and mesenteric lymph node	[95,96]
**CD4^+^ CD8αα^+^ T cell**	SI epithelium	Tolerance to dietary antigens	?	Promoted by dietary tryptophan & indole derivatives ↑ in SI after weaning	↓ in SI	↓ in SI	[97]
**Regulatory T cell**	Lamina propria, Peyer’s patches, mesenteric lymph node	Intestinal inflammation suppression through IL-10 and TGFβ production Tolerance to dietary antigens	FAO, TCA, OXPHOS	High fibre promotes Treg ↑ in colon and SI after weaning	↓ in colon; ↓ or similar in SI, Peyer’s patch; inconsistent in mesenteric lymph node ↓ in RORγt Treg (peripheral induced Treg) in colon while Helios^+^ thymic ones mostly similar	↓ in colon and Peyer’s patch; similar in mesenteric lymph node; ↓ in RORγt Treg in colon, SI and mesenteric lymph node; ↓ or similar Helios^+^ and GATA3^+^ Treg in colon	[98,99,100]
**γδ T cell**	Lamina propria, Peyer’s patches, mesenteric lymph node, Intestinal epithelium	Intestinal inflammation suppression through IL10 and TGFβ production Epithelial repair and protection through KGF-1 and IL-17 production Host-microbiota mutualism maintenance through antimicrobials production scavenger receptor 2 (SCART-2) positive γδT cells produce IL-17 in the colon of mice to control antimicrobial epithelial responses	IFNγ-producing γδ T cells: glycolysis IL-17 producing γδ T cells: TCA OXPHOS		Similar IL-17 production in SI, ↓ antimicrobials production in SI	Similar or ↑ in SI; ↓ IL-17 production in SI; ↓ antimicrobials production in SI Minor impact on gut intraepithelial lymphocyte population	[101,102,103,104,105,106]
**NK T cell**	Lamina propria, Intestinal epithelium	Defence against microbial pathogen Host-microbiota mutualism Th1 and Th2 cytokines production IL-17 and IL-22 production	Generally, blocking glycolysis results in NKT activation		Similar in mesenteric lymph node and Peyer’s patch; ↑ in colon	↑ in colon	[107,108,109,110]
**NK cell**	GALT, mesenteric lymph node, intestinal epithelium, lamina propria	Cytotoxicity, IL-22 production to modulate epithelial survival and remodelling, IFN-γ production	Resting NK cells: OXPHOS Activated NK cells: glycolysis	↓ in SI after weaning ↓ NK cell activity after weaning	Similar in Peyer’s patch, and mesenteric lymph node	RORγt^neg–int^NK1.1^high^ similar, but RORγt^high^NK1.1^int^ ↓ in lamina propria ↓ IL-22 production	[97,111,112]
**Dendritic cell**	GALT, mesenteric lymph node, intestinal epithelium, lamina propria	Antigen sampling and presenting Gut tropism imprinting Regulatory and effector T cell induction Food/oral/microbiota tolerance	Tolerogenic DC: OXPHOS Immunogenic DC: glycolysis		↓ in SI, colon and mesenteric lymph node	↓ or similar in mesenteric lymph node ↓ type I IFN production ↓ or similar in colon ↓ in SI CX_3_CR1^+^ DC ↓ in SI CD103^+^ DC ↓ in mesenteric lymph node	[113,114,115,116]
**Macrophage**	GALT, mesenteric lymph node, intestinal epithelium, lamina propria	Intestinal inflammation suppression through IL10 production Host-microbiota mutualism Apoptotic or damaged cell cleavage Treg vs Th17 balance	M1 pro-inflammatory: highly glycolytic, fatty acid synthesis, reduced TCA cycle M2 anti-inflammatory: FAO, OXPHOS, decreased glycolysis	Yolk sac/fetal liver-derived Macrophage would be diluted after weaning by accumulation of circulating Ly6C^high^ monocytes	↓ in SI, and colon, similar in Peyer’s patches, and mesenteric lymph node	↓ in colon	[117,118,119]
**Neutrophil**		Antimicrobials production, host-microbiota mutualism, immune cell activation and recruitment, mucosal/epithelial repairing	Immature, c-Kit+ neutrophils: OXPHOS Mature neutrophils: glycolysis				[31,120]
**Innate lymphoid cell**	Intestinal epithelium, mucosal surface	*Type 1 innate lymphoid cells (ILC1):* IFNγ and TNFα production, anti-virus, -cancer, -intracellular pathogen responses *Type 2 innate lymphoid cells (ILC2):* IL5 and IL13 production, anti-helminth responses, tissue repairing *Type 3 innate lymphoid cells (ILC3):* IL17 and IL22 production, intestinal lymphoid organ development, host-microbiota mutualism and host defence	Resting ILC1: ? Active ILC1: ? resting ILC2s: OXPHOS or FAO Active ILCs: glycolysis and high rates of OXPHOS Resting ILC3: ? suggested glycolysis- still largely unclear Active ILC3: FAO and synthesis but unclear	ILC3: LTi cells developed in fetus would be diluted by post-birth develop LTi-like ILC3 after weaning in intestinal lamina propria	↓ ILC3 and ILC1 in Peyer’s patch; ↑ ILC3 in terminal ileum Peyer’s patch; ↓ GM-CSF^+^ ILC3 in colon ILC1 and ILC2 expression become ILC3-like	Similar ILC1 in SI; ↑ ILC2 in SI; similar or ↑ ILC3 in SI	[111,121,122,123,124,125]
**Goblet cell**	SI, colon epithelium	Mucus secretion	OXPHOS is necessary for goblet cell differentiation	Weaning effect in pigs High protein diet increases goblet cell number and promotes mucus secretion in ileum and alter goblet cell distribution in colon	Similar in colon ↓ in SI ↓ mucus secretion in colon upon Metronidazole treatment	Similar in colon	[126,127,128,129,130]
**Paneth cell**	SI (crypt)+++, colon+	Antimicrobial peptides and cytokines production Stem cell niche support	Highly glycolytic		↓ in SI Distorted crypts and fewer granules in cells	↓ in SI ↓ Reg3γ expression in SI	[131]
**Tuft cell**	Intestinal epithelium	Chemosensation, IL-25 production to promote ILC2 expansion, anti-parasitic responses	?	↑ in SI and colon upon fasting and refeeding ↑ after weaning ↑ in SI after succinate feeding		Similar in colon	[58,132,133]
**Microfold cell**	GALT, Peyer’s patch, mesenteric lymph node,	(Antigen) Transportation through trans-cellular endocytosis, cytokines/costimulatory signal (IL1) secretion Antigen uptake	?			Similar in Peyer’s patch	[134]

↑ = Increased, ↓ = Decreased ? = Unknown. Abbreviations: SI small-intestine, STAT3 Signal transducer and activator of transcription 3, Th1 T helper 1, Th2 T helper 2, Th17 T helper 17, TCR T cell receptor, FAO fatty acid oxidation, IFNγ interferon-gamma, TGFβ transforming growth factor beta, RORγt Retinoic acid-related orphan receptor gamma t, IL-17 interleukin-17, IL-22 interleukin-22, NKT natural killer T cell, GALT gut-associated lymphoid tissue, NK natural killer cell, DC dendritic cell, ILC1 group 1 innate lymphoid cells, ILC2 group 2 innate lymphoid cells, ILC3 group 3 innate lymphoid cells, LTi lymphoid tissue inducer, IL-25 interleukin-25.

**Table 2 nutrients-13-00823-t002:** Summary of common food additives and immune impacts.

Additive	Immune Effects	Reference
Citrate	Promoted adipose tissue inflammation (IL-6, TNF, IL-1β) and insulin resistance when orally administered in combination with a high sucrose diet in mice.	[166]
	High levels potentiated inflammatory response in LPS- stimulated THP-1 cells *in vitro.*	[168]
Emulsifiers: Polysorbate-80 and Carboxymethylcellulose (CMC)	Both P80 and CMC promoted low grade intestinal inflammation (increased MPO, CXCL1, CXCL2 and TNF expression), exacerbating colitis and carcinogenesis in an AOM/DSS mouse model.	[181]
	Heightened low grade inflammation, induced via microbial changes, resulting in metabolic syndrome and IBD.	[164]
	CMC promoted bacterial overgrowth within the small intestine of in IL-10 deficient mice (a model for spontaneous colitis). Did not show significant difference in intestinal inflammation.	[182]
Emulsifiers: Maltodextrin (MDX)	Induced upregulation of genes involved in lipid and carbohydrate metabolism in intestinal epithelial cells in vivo. Induced ER stress in intestinal epithelial cells, greater TNF and IL-1β expression and worse DSS colitis. Inhibition of endoplasmic reticulum (ER) stress in maltodextrin fed mice protected from worsened colitis. This effect was ameliorated with ER stress inhibition.	[183]
Sodium chloride	Activated p38/MAPK pathway during cytokine induced Th17 cell polarisation, leading to Th17 bias. Th17 cells polarised in high NaCl conditions produced more GM-CSF, TNF and IL-2. This led to more severe experimental autoimmune encephalomyelitis (EAE) in high salt intake mice.	[184]
	Worsened pathology in the IL-10^-/-^ mouse model of colitis. Elevated colonic expression of TNF, IL-12β, IL-23α, IL-1β. Mice infected with *Salmonella typhimurium* had greater IL-17A, IL-12α, IL-12β, TNF, IL-23α, IL-6 and IFNγ mRNA expression on high sodium diet.	[173]
	A small human study found activation of CD14^++^ monocytes and expansion of “intermediate” CD14^++^CD16^+^ monocyte populations in the peripheral blood. *In vitro* follow up studies demonstrated ROS activation of CD14^++^ monocytes in high sodium media.	[185]
	*In vitro* high NaCl media supressed the induction of AKT and mTOR signalling and increased oxidative phosphorylation and glycolysis in macrophages. High NaCl also dampened the ability of M2 macrophages to supress CD4^+^ and CD8^+^ effector T cell proliferation in vitro but enhanced *Nos2* expression in M1 macrophages when stimulated with LPS. Impaired wound healing in mice.	[171]
	A high NaCl environment induced IL-1β secretion in bone marrow derived macrophages in vitro via through NLRP3 inflammasome and subsequent caspase-1 activation. Mice on a high NaCl diet had an enhanced IL-17 production when immunised with OVA and LPS. This was caspase-1 specific.	[174]
	High NaCl blocked the suppressive capacity of human Tregs *in vitro.* Tregs in high salt fed mice switched to an IFNγ producing phenotype and similarly lost suppressive function. Treg suppressor function was regained by blocking IFNγ or by silencing serum and glucocorticoid-regulated kinase 1 (SGK1) in Tregs.	[186]
	High salt in vivo induced IL-17 like Tregs in the thymus and promoted the generation of peripheral RORγt^+^ iTregs in Th17 polarising conditions in an SGK1 dependant manner.	[187]
	Mice fed a high salt diet for 1 week had impaired neutrophil function and more severe pyelonephritis	[188]
	Worsened EAE symptoms in mice with high NaCl intake. Enhanced expression of TNF, IL-12, IL-1β, iNOS and IL-6 in the CNS of EAE mice. Promoted pro-inflammatory M1 macrophage activation and activation of NF-κB signalling.	[172]
Sucralose	Glucose intolerance via actions on microbiome.	[189]
	Promoted a pro-inflammatory environment in the liver as measured by TNF and iNOS transcription. Also altered microbiome composition and microbial metabolic profile to one which is associated with inflammation.	[190]
Titanium dioxide (E171)	Promoted macrophages to produce TNF and IL-10 ex-vivo in the presence of LPS.	[191]
	Increased macrophage and CD8^+^ T cells in the colon of orally exposed mice.	[178]
	Long term exposure led to spontaneous preneoplastic development in the small intestine of rats. Also resulted in increased IL-17 and IFNγ production when cells from Peyer’s patches were restimulated in vitro and increased IL-10, TNF-a, IL8, IL-6, IL-1β and IFNγ in colonic mucosa or orally exposed mice.	[179]
	Chronic exposure worsened DSS- induced psoriasis via activation of NLRP3 inflammasome	[192]

Abbreviations: MPO Myeloperoxidase, CXCL1 chemokine (C-X-C motif) ligand 1, CXCL2 chemokine (C-X-C motif) ligand 2, iNOS inducible nitric oxide synthase.

## Data Availability

Not applicable.

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
