# Peer review of "How Changes in the Nutritional Landscape Shape Gut Immunometabolism"

_nutrients, 2021, doi:10.3390/nu13030823_

Round 1

Reviewer 1 Report

It's definetely an overall good study. I find very important the analysis on how the nutrients and the metabolism affect the defence, immune response, gut homeostasis - that would probably explain in the near future how the diet affects the intestinal pathology (inflammatory, cancer).

The idea is scientifically sound and, even if a large number of publications are dealing with similar subjects, quite original. The abstract reflects the main article and the aim of the work is clear and given.  The introduction is giving the idea of what follows and it is nice and relevant. Material and methods are extensive, clear, simple and easy to follow. The full range of the main diet substances has been reported and analysed as the main cell types influenced by the diet components. Nonetheless, the effect of the byproducts of the gut metabolism is also analysed and reported. The immunometabolism has been thoroughly explained in a very clear and concise manner. The results are expected and reliable, the limitations stated and the references relevant.

I honestly have nothing to suggest.

Author Response

We thank the reviewer for their review of our manuscript as well as their positive feedback.

Reviewer 2 Report

The topic is interesting,a comprehensive review.

Author Response

(The authors gave the same response as above.)

Reviewer 3 Report

The manuscript submitted by Tan and colleagues provides and in-depth review of how dietary components affect gut immunemetabolism. The review is well-written, comprehensive, and the figures are great.

Major comments

  • None

Minor comments

  • Please add a reference to line 68 – “For example, short-chain fatty acids produced under high fibre feeding condition can bind to specific GPCR expressed on immune cell to alter their phenotype.”

Author Response

We thank the reviewer for their review of our manuscript as well as their positive feedback. We will incorporate the existing reference (55) to line 68 as suggested.